# Systematic review of infant and young child feeding practices in conflict areas: what the evidence advocates

Amna Rabbani,[1] Zahra A Padhani ,[1] Faareha A Siddiqui,[1] Jai K Das ,[1] Zulfiqar Bhutta [2]

¹Division of Women and Child Health, Aga Khan University, Karachi, Sindh, Pakistan
²Centre for Global Child Health, The Hospital for Sick Children, Toronto, Ontario, Canada

**Correspondence to**
Professor Zulfiqar Bhutta;
zulfiqar.bhutta@sickkids.ca

## ABSTRACT

**Background** Breast feeding in conflict settings is known to be the safest way to protect infant and young children from malnourishment and increased risk of infections. This systematic review assesses the evidence on infant and young child feeding (IYCF) practices in conflict settings.

**Methodology** We conducted a search in PubMed and CENTRAL and also searched for grey literature from the year 1980 to August 2019. We included studies conducted in settings inflicted with armed conflict; which comprised settings undergoing conflict, as well as, those within 5 years of its cessation. Studies were included if they discussed IYCF practices, barriers, programmes and guidelines to promote and improve IYCF practices. Two review authors independently evaluated and screened studies for eligibility and extracted data; followed by a descriptive and thematic analysis.

**Results** We included 56 studies in our review including 11 published articles and 45 reports from grey literature and broadly classified into four predetermined sections: epidemiology (n=24), barriers/enablers (n=18), programmes/interventions (n=15) and implementation guidelines (n=30). Epidemiological evidence shows that IYCF practices were generally poor in conflict settings with median prevalence of exclusive breast feeding at 25%, continued breast feeding at 29%, bottle feeding at 58.3%, introduction to solid, semisolid or soft foods at 71.1% and minimum dietary diversity at 60.3%.

IYCF practices were affected by displacement, stress, maternal malnutrition and mental health, family casualties and free distribution of breast milk substitutes. To improve IYCF, several interventions were implemented; including, training of health workers, educating mothers, community networking and mobilisation, lactation-support service, baby friendly hospital initiative, mother–baby friendly spaces and support groups.

**Conclusion** The evidence suggests that IYCF practices are generally poor in conflict inflicted settings. However, there is potential for improvement by designing effective interventions, responsibly disseminating, monitoring and implementing IYCF guidelines as prescribed by WHO development partners, government and non-government organisations with dedicated funds and investing in capacity development.

## INTRODUCTION

Optimal infant and young child feeding (IYCF) practices play a critical role in determining the nutritional status, health, growth

### Strengths and limitations of this study

► To our knowledge, this is the first systematic review on infant and young child feeding (IYCF) practices in conflict settings that looks at the evidence on the current practices of breast feeding and complementary feeding, and assesses specific barriers to adapting optimal IYCF practices. This review also explores the evidence on effective strategies to improve IYCF practices, as well as evidence from implementation guidelines, which suggest the way IYCF should be approached in conflict contexts.

► The review highlights the suboptimal IYCF practices in children affected by conflict.

► The review highlights the evidence derived from strategies/interventions implemented in conflict areas to improve IYCF practices, although weak, provides important insights for future approaches to improve IYCF practices.

► The review is limited by the scarcity of evidence as many programmes in conflict contexts are not reported. And even within published reports, very few studies have clearly specified the scale of the intervention, year of conflict, year of 'baseline' data or performed a formal evaluation of the programme and its impact on IYCF outcomes.

and development of children, along with improving the health of mothers.[1-4] The current guidelines suggest breast feeding should to be initiated within the first hour of birth and infants be exclusively breast fed for the first 6 months of life, that is, receive only breast milk, with the exception of oral rehydration syrups solutions, drops/of vitamins, minerals and medicines.[1 5 6] Exclusive breast feeding (EBF) offers the required nourishments for normal growth and development till 6 months of age[2]; thereafter safe, timely and nutritionally adequate complementary foods should be added to the diet of infants, along with continued breast feeding up to 2 years of age.[1 5 6]

Children who have been breast fed for longer periods of time tend to exhibit lower

odds of infectious morbidity and mortality,[7] as infants who are not breast fed have sixfold greater risk of infections related in the first 2 months of life when compared with infants that have been adequately breast fed.[8] The current evidence suggests that high-income countries (HICs) practice shorter duration of breast feeding (<20%) compared with low/middle-income countries (LMICs).[7] However, even within LMICs, approximately only 37% of infants younger than 6 months are exclusively breast fed.[7] Just scaling up and promoting breast feeding to a universal level could possibly prevent 823 000 annual deaths in children under the age of 5 years[7] and 13.8% of these under 2 years of age.[7] After 6 months of age, energy–and–nutrient dense foods that can be easily eaten and digested should be added to infants' diet in order meet their dietary demands.[9] Both breast feeding and appropriate complementary feeding are pivotal for child growth and the prevention of disease and malnutrition.[1] Breast feeding coupled with complementary feeding has the potential to reduce mortality among children under the age of 5 years by 19%.[5]

During times of armed conflict, vulnerable groups including children bear the greatest negative consequences. The onset of conflict increases death rates by up to 24 times, with adverse effects especially for children under the age of 5 years.[10] Newborns are specifically at a higher risk of dying if they are poor, exposed to unsafe environments or if they are within a conflict setting.[8] Armed conflict significantly impacts breast feeding practices with lower rates of breast feeding observed in war-torn areas.[11] Before Lebanon's conflict in 2006, approximately 27% of mothers exclusively breast fed for the first 4 months of life and after the conflict escalated, the breast feeding practices were severely affected with most mothers discontinuing breast feeding altogether or initiating mixed feeding and/or reduced breast feeding.[12] Similarly, in these war-torn areas, complementary feeding may also become severely eroded and disrupted.[1] This could be attributed to safety, access to sufficient quantity and quality of complementary foods, and also appropriate knowledge of complementary feeding.[1] Often, misconceptions associated with the introduction of solid food results in mothers or caregivers, results in either initiating solid food early or waiting more than required.[13] Suboptimal complementary food intake can result in deterioration of health status of infants and young children, increasing the risk of morbidity and mortality.[1]

In conflict settings; breast feeding and appropriate complementary feeding is of crucial importance because it is regarded as the safest way to protect infants and young children from infections and malnourishment.[14] This can be corroborated by the fact that during emergency situations; mortality rates of artificially fed infants are greatly elevated in comparison to breastfed babies,[8] as the risk mortality due to diarrhoea and other infectious diseases are 20 times higher than infants who have been exclusively breast fed.[12] This is due to prevalent unhygienic conditions coupled with lack of safe water and facilities

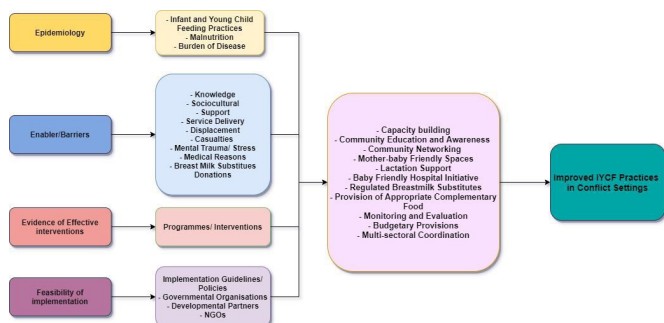

**Figure 1** Conceptual framework. IYCF, infant and young child feeding; NGO, non-governmental organisation.

to sterilise feeding bottles and prepare formula safely. Support for optimal breast feeding, re-lactation, and timely introduction of complementary foods should be the first choice of intervention during conflict situations to mitigate feeding problems for infants and young children[9] and should not be undermined by inappropriate distribution of breast milk substitutes (BMS).[2 9 10] There is currently no existing comprehensive review for IYCF in conflict settings. The objective of this systematic review is to assess the evidence on IYCF practices, factors associated with IYCF, evidence on the interventions undertaken and guidelines to improve IYCF practices in children under 2 years of age living in conflict settings.

## CONCEPTUAL FRAMEWORK

During armed conflicts, IYCF practices are disrupted due to displacement, casualties, lack of resources, stress and malnutrition, as well as due to unregulated BMS donations by agencies. We developed a conceptual framework to guide this review and highlighting the IYCF practices and factors responsible for improving IYCF practices in an armed conflict and post-conflict settings (figure 1). It also focuses on the evidence of current IYCF practices, factors associated with improving IYCF and the evidence from interventions and programmes implemented in these settings. We also explored existing implementation guidelines and programme recommendations for improving IYCF practices from different agencies working in such settings.

## METHODOLOGY

We conducted a systematic review for available published and grey literature, assessing four domains including: epidemiology (coverage of key IYCF and malnutrition indicators), enablers/barriers (for recommended IYCF practices), interventions/programmes (effectiveness in improving IYCF practices) and implementation guidelines to improve IYCF practices in conflict settings.

### Eligibility criteria

We included studies assessing the impact of conflict on IYCF practices and enablers/barriers, interventions/ programmes and guidelines to promote or facilitate IYCF

## Box 1  Search strategy

("Disasters"[Mesh] OR "Disaster Victims"[Mesh] OR "Natural Disasters"[Mesh] OR mass disaster OR catastrophe* OR crisis OR crises OR war OR "afghan campaign" OR "Armed Conflicts"[Mesh] OR conflict OR conflicts OR war OR avalanche* OR cyclone* OR drought* OR earthquake* OR flood* OR hurricane* OR landslid* OR land slide OR land slides OR mudslid* OR mudslides OR mud slide OR mud slides OR storm* OR tornado* OR tsunami* OR typhoon* OR volcano*) AND ("Refugees"[Mesh] OR refugee OR asylum seeker* OR returnees OR "IDP" OR "IDPs" OR internally displaced person* OR mother* OR "lactating" OR "lactating mother" OR "lactating mothers") AND (Breastfeed* OR "breast feeding" OR "breastfed" OR "exclusive breastfeeding" OR "human milk" OR "lactation" OR infant feed* OR complementary feed* OR "IYCF" OR "Infant and Young Child Feeding" OR "early initiation" OR "colostrum" OR "breastfeeding practices" OR "breastfeeding promotion" OR "breastfeeding facilitation" OR barriers OR "breast milk collection" OR "breast milk collections" OR "breast milk expression" OR "breast milk expressions" OR "breast pumping" OR "breast pumpings" OR "breastmilk collection" OR "breastmilk collections" OR "breastmilk expression" OR "breastmilk expressions" OR "milk bank" OR "feeding bank").

in conflict settings. We included studies conducted in an armed conflict setting, defined as 'a political conflict in which armed combat involves the armed forces of at least one state (or one or more armed factions seeking to gain control of all or part of the state) and in which people have been killed by the fighting during the course of the conflict'.[15] We included studies conducted during conflict and within 5 years of its cessation. Included studies comprised primary research articles, reports and grey literature and policy and guidelines. We included all studies conducted in conflict settings of low, low-middle and upper-middle income countries classified by World Bank,[16] during the period of 1980–2018. Additionally, studies conducted in refugee camps of HIC were also included for analysis.

We excluded studies conducted in HICs apart from refugee camps. Clinical studies (exclusively looking at microbiological/laboratory outcomes/screening or diagnostic test evaluations or surgical techniques/outcomes), mathematical modelling or economic studies (with no empirical data/information), systematic and literature reviews were excluded from the study. We also excluded studies conducted in humanitarian emergencies apart from armed conflict.

### Data management
We searched PubMed and CENTRAL using our search strategy constructed by using Medical Subject Headings (MeSH) and key words (box 1). The bibliographies of relevant systematic reviews and included studies were searched to identify the missing records in the database search. We only included articles in English language. Additionally, a grey literature search was conducted on Google and websites and publications of relevant agencies such as WHO, UNICEF, United Nations High Commissioner for Refugees (UNHCR), International Baby Food Action Network, Emergency Nutrition Network, Baby Milk Action, Save the Children, Action Against Hunger and Wellstart International. We also searched for additional data by entering titles of all included studies on Google Scholar and reviewing the first 10 pages to include any relevant missing studies.

After running the electronic search, all records were imported to EndNote software[17] and were de-duplicated prior to title and abstract screening. Two reviewers independently screened titles and abstracts, followed by full text screening. All the discrepancies were resolved after discussion at each stage and a third reviewer was contacted in case two reviewers were unable to reach a consensus. Data were extracted independently by two reviewers from the included studies in an excel sheet after the full text screening. We calculated the median of all IYCF and malnutrition indicators from the studies identified and conducted a descriptive and thematic analysis of included studies to explore and synthesise information on the contextual factors and intervention and recommended implementation strategies.

### Patient and public involvement
This research was done without patient involvement. Patients were not invited to contribute to the writing or editing of this document for readability or accuracy.

## RESULTS
We identified a total of 56 studies (figure 2) which were broadly classified into the pre-determined four sections; epidemiology, enablers/barriers, programmes/interventions and implementation guidelines.

### Epidemiology
Twenty-four studies reported on the prevalence of breast feeding and the burden of disease in different conflict-affected regions.[10 13 14 18–38] Eighteen were reports,[10 13 14 18–24 26 28–30 32 33 37 38] five were cross sectional studies[25 27 34–36] and one was a cohort study.[31] Seven studies were conducted in Middle East,[13 14 19 21–23 26] six in Europe,[18 19 27 30 33 35] five in Africa,[18 20 25 31 32] four in Asia[24 28 29 38] and two reported on prevalence in more than one region which included Africa[10], Asia[10 37] and Middle East.[37] The included studies collected data through various sources and methods; five studies included data from national assessments done by the WHO and other developmental partners,[10 19 21 22 37] five conducted cross sectional household surveys,[26 27 33 34 36] four collected data from multiple indicator cluster survey,[13 14 29 35] four conducted SMART surveys,[18 20 28 38] one distributed questionnaires,[25] one collected data from national surveillance system registry,[31] one through Disease Early Warning System surveillance network[24] and two did not specify.[23 32]

Twenty studies reported on IYCF according to the WHO IYCF indicators which are summarised in table 1.[13 18–21 23 25–38] These studies analysed IYCF indicators divided into WHO eight core indicators (early initiation

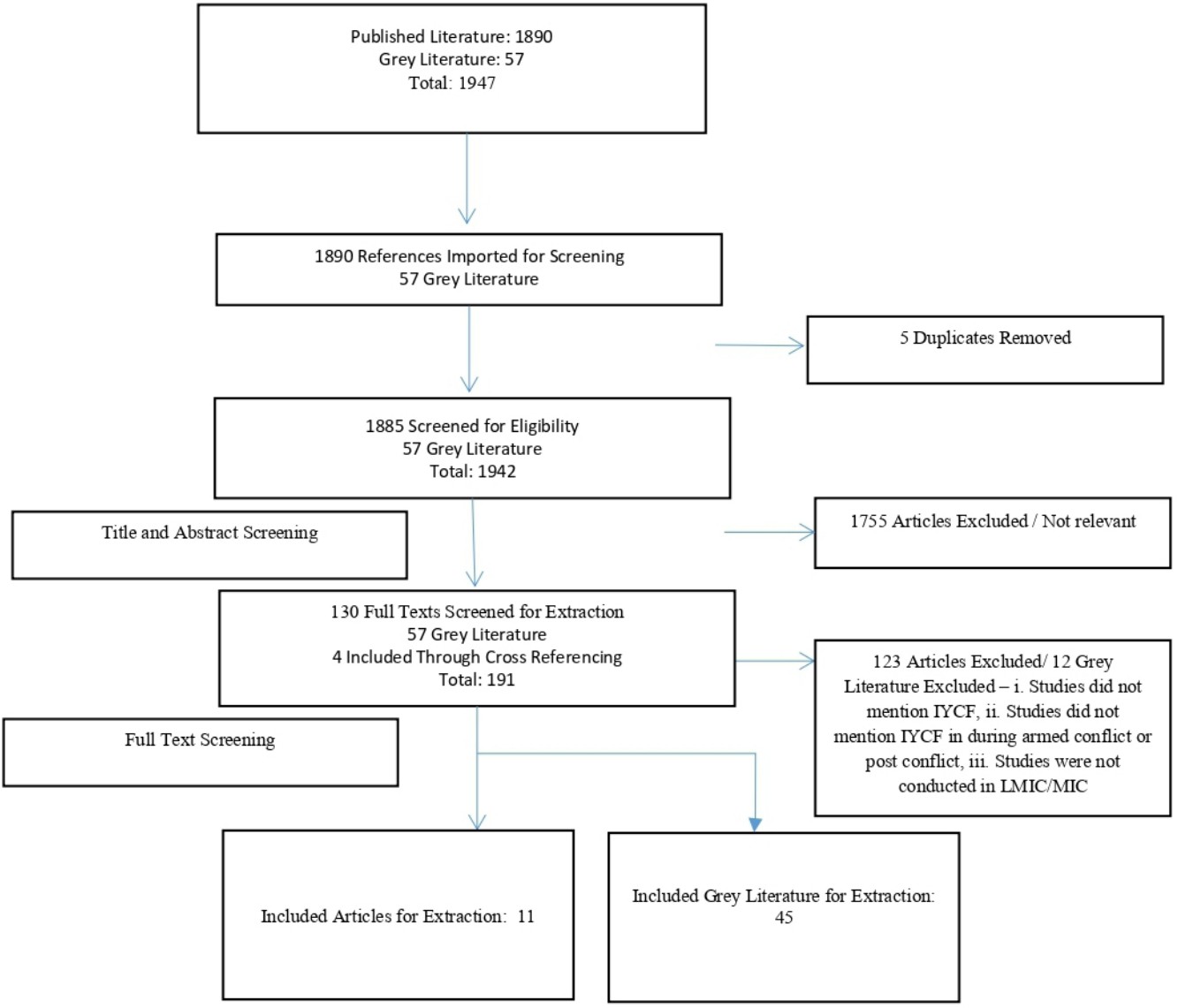

**Figure 2** Search flow diagram. IYCF, infant and young child feeding; LMIC, low/middle-income countries; MIC, middle-income countries.

of breast feeding, EBF under 6 months, continued breast feeding at 1 year, introduction of solid, semisolid or soft foods, minimum dietary diversity, minimum meal frequency, minimum acceptable diet and consumption of iron-rich or iron-fortified foods) and seven optional indicators (children ever breast fed, continued breast feeding at 2 years, age-appropriate breast feeding, predominant breast feeding under 6 months, duration of breast feeding, bottle feeding and milk feeding frequency of non-breastfed children).[39] Using the data extracted from included studies, the median prevalence of early initiation of breast feeding in conflict-affected areas was 51% (range: 31.3%–85%), EBF was 25% (range: 5.5%–77.1%), proportion of children with appropriate introduction of solid, semisolid or soft foods was 71.1% (range: 40.7%–98.6%), proportion of children with minimum dietary diversity was 60.3% (range: 9.2%–79.4%), children ever breast fed was 92% (range: 62.8%–98.4%), continued

breast feeding at 2 years was 29% (range: 9%–66%), age-appropriate breast feeding was 43.2% (range: 19.5%–77.8%), predominant breast feeding was 31.3% (range: 7.1%–77.3%) and bottle feeding was 58.3% (range: 31.8%–71.4%).

Two studies[25 27] reported on association of breast feeding with malnutrition in displaced Bosnian[27] and Saharawi children.[25] Infants were more likely to be malnourished who were never breast fed (OR: 1.78, 95% CI 1.26 to 2.52) or who were not breast fed for at least 4 months (OR: 1.45, 95% CI 1.02 to 2.07) than those who were ever breast fed or were breast fed for 4 months. Malnutrition persisted among infants who were not exclusively breast fed and infants who were breast fed for less than 5–6 months were more likely to be malnourished (OR: 1.98, 95% CI 1.01 to 7.35) than those who were breast fed for more than 6 months.[27] After adjusting for diseases, mother's body mass index and child's age; prevalence of underweight

**Table 1** IYCF practices in conflict settings

| Indicators | Target population: country | Setting | Estimates median (range) | Sample size median (range) | Scale |
|---|---|---|---|---|---|
| **Core indicators** | | | | | |
| Early initiation of breast feeding (n=10)[18 20 21 25 26 28 32 35 36 38] | ▲ IDPs: South Sudan, Ukraine<br>▲ Refugees: Algeria, Jordan, Kenya, Lebanon, Pakistan<br>▲ Not specified: Bosnia-Herzegovina, CAR, Yemen | ▲ Camp: Algeria, Jordan, Kenya, Lebanon<br>▲ Community: Pakistan, South Sudan, Ukraine, Yemen | 51%<br>(range: 31.3%–85%) | 357<br>(range: 111–1368) | ▲ Camps: Algeria, Jordan, Kenya<br>▲ Village: Pakistan<br>▲ District: Yemen<br>▲ National: CAR |
| Exclusive breast feeding under 6 months (n=16)[10 13 20 21 25 26 28-30 32-38] | ▲ IDPs: Bosnia-Herzegovina, Ukraine, Sierra Leone, South Sudan<br>▲ Refugees: Algeria, Greece, Jordan, Kenya, Lebanon, Pakistan<br>▲ Not specified: Afghanistan, Bosnia-Herzegovina, Iraq, Kosovo, Lebanon, Syria, Yemen | ▲ Camp: Algeria, Kenya, Lebanon<br>▲ Community: Bosnia-Herzegovina, Iraq, Kosovo, Pakistan, South Sudan, Ukraine, Yemen<br>▲ Community and healthcare facility: Jordan | 25%<br>(range: 5.5%–77.1%) | 432<br>(range: 58–250000) | ▲ Camps: Algeria, Jordan, Kenya<br>▲ Village: Pakistan<br>▲ District: Yemen<br>▲ National: Kosovo |
| Continued breast feeding at 1 year (n=7)[21 28 32 33 35 36 38] | ▲ IDPs: Ukraine<br>▲ Refugees: Greece, Kenya, Lebanon, Pakistan<br>▲ Not specified: Bosnia-Herzegovina, Yemen | ▲ Camp: Kenya, Lebanon<br>▲ Community: Pakistan, Ukraine, Yemen | 57.20%<br>(range: 8.5%–78.2%) | 326<br>(range: 38–1368) | ▲ Camp: Kenya<br>▲ Village: Pakistan<br>▲ District: Yemen |
| Introduction of solid, semisolid or soft food (n=6)[25 32 33 35 36 38] | ▲ IDPs: Ukraine<br>▲ Refugees: Algeria, Greece, Kenya, Pakistan<br>▲ Not specified: Bosnia-Herzegovina | ▲ Camp: Algeria, Kenya<br>▲ Community: Pakistan, Ukraine | 71.05%<br>(range: 40.7%–98.6%) | 477<br>(range: 111–1368) | ▲ Camps: Algeria, Kenya<br>▲ Village: Pakistan |
| Minimum dietary diversity (n=5)[18 20 28 33 38] | ▲ IDPs: South Sudan<br>▲ Refugees: Greece, Pakistan<br>▲ Not specified: CAR, Yemen | ▲ Community: Pakistan, South Sudan, Yemen | 60.25%<br>(range: 9.2%–79.4%) | 309<br>(range: 148–1055) | ▲ Village: Pakistan<br>▲ District: Yemen<br>▲ National: CAR |

Continued

**Table 1** Continued

| Indicators | Target population: country | Setting | Estimates median (range) | Sample size median (range) | Scale |
|---|---|---|---|---|---|
| Minimum meal frequency (n=6)[18 20 28 33 36 38] | ▲ IDPs: South Sudan, Ukraine ▲ Refugees: Greece, Pakistan ▲ Not specified: CAR, Yemen | ▲ Community: Pakistan, South Sudan, Ukraine, Yemen | 58% (range: 58%–97.6%) | 432 (range: 148–1055) | ▲ Village: Pakistan ▲ District: Yemen ▲ National: CAR |
| Minimum acceptable diet (n=5)[18 20 28 33 38] | ▲ IDPs: South Sudan ▲ Refugees: Greece, Pakistan ▲ Not specified: CAR, Yemen | ▲ Community: Pakistan, South Sudan, Yemen | 24.95% (range: 30.5%–33%) | 309 (range: 148–1055) | ▲ Village: Pakistan ▲ District: Yemen ▲ National: CAR |
| Consumption of iron rich and iron fortified food (n=2)[26 36] | ▲ IDPs: Ukraine ▲ Refugees: Jordan | ▲ Camp: Jordan ▲ Community: Ukraine | 51.61% (range: 29%–84.7%) | 379 (range: 281–477) | ▲ Camp: Jordan |
| **Optional indicators** | | | | | |
| Children ever breast fed (n=5)[27 30 33 36 38] | ▲ All (refugees, displaced and not displaced): Bosnia-Herzegovina ▲ IDPs: Ukraine ▲ Refugees: Greece, Macedonia, Pakistan | ▲ Camp: Macedonia ▲ Community: Bosnia-Herzegovina, Pakistan, Ukraine | 92% (range: 62.8%–98.4%) | 766 (range: 148–1123) | ▲ Camp: Macedonia ▲ Village: Pakistan ▲ National: Bosnia-Herzegovina |
| Continued breast feeding at 2 years (n=6)[21 29 33 35 36 38] | ▲ IDPs: Ukraine ▲ Refugees: Lebanon, Pakistan, Greece ▲ Not specified: Bosnia-Herzegovina, Iraq | ▲ Camp: Lebanon ▲ Community: Iraq, Pakistan, Ukraine | 29% (range: 9%–66%) | 477 (range: 148–55 194) | ▲ Village: Pakistan ▲ National: Iraq |
| Age appropriate breast feeding (n=3)[33 35 38] | ▲ Refugees: Greece, Pakistan ▲ Not specified: Bosnia-Herzegovina | ▲ Community: Pakistan | 43.20% (range: 19.5%–77.8%) | 602 (range: 148–1055) | ▲ Village: Pakistan |
| Predominant breast feeding under 6 months(n=5)[25 30 33 35 38] | ▲ Refugees: Algeria, Greece, Pakistan ▲ Not specified: Bosnia-Herzegovina, Kosovo | ▲ Camp: Algeria ▲ Community: Kosovo, Pakistan | 31.30% (range: 7.1%–77.30%) | 176 (range: 111–1055) | ▲ Village: Pakistan |

Continued

**Table 1** Continued

| Indicators | Target population: country | Setting | Estimates median (range) | Sample size median (range) | Scale |
|---|---|---|---|---|---|
| Duration of breast feeding (n=2)[27 31] | ▲ All (refugees, displaced and not displaced): Bosnia- Herzegovina<br>▲ IDPs: Guinea-Bissau | ▲ Community: Bosnia-Herzegovina, Guinea-Bissau | 22.7 months | 2149 (range: 1741–2556) | ▲ City: Guinea-Bissau<br>▲ National: Bosnia-Herzegovina |
| Bottle feeding (n=3)[21 36 38] | ▲ IDPs: Ukraine<br>▲ Refugees: Lebanon, Pakistan | ▲ Camp: Lebanon<br>▲ Community: Pakistan, Ukraine | 58.30% (range: 31.8%–71.4%) | 477 (range: 174–1055) | ▲ Village: Pakistan |
| Milk feeding frequency of non-breastfed children (n=1)[33] | ▲ Refugees: Greece | ▲ Not stated | 33.90% | 148 | ▲ Not stated |
| **Malnutrition** | | | | | |
| Underweight (n=4)[10 25 28 38] | ▲ IDPs: Afghanistan, Sierra Leone<br>▲ Refugees: Algeria, Pakistan<br>▲ Not specified: East Timor, Yemen | ▲ Camp: Algeria<br>▲ Community: Pakistan, Yemen | 33.10% (range: 12.01%–48%) | 1055 (range: 111–250000) | ▲ Camp: Algeria<br>▲ Village: Pakistan<br>▲ District: Yemen |
| Acute malnutrition (n=3)[10 24 36] | ▲ All ((IDPs, refugees and residents): Southern Somalia<br>▲ IDPs: Ukraine, Pakistan | ▲ Camp: Southern Somalia, Ukraine<br>▲ Camps, hospitals and mobile clinics: Pakistan | 30.10% (range: 0.5%–81%) | 80 000 (range: 477–3 000 000) | ▲ Camps and healthcare facilities: Pakistan |
| GAM (n=7)[14 18 20 23 28 32 38] | ▲ IDPs: South Sudan<br>▲ Refugees: Jordan, Kenya, Pakistan<br>▲ Not specified: CAR, Yemen | ▲ Camp: Jordan, Kenya<br>▲ Community: Pakistan, South Sudan, Yemen | 9.95% (range: 2.6%–25.1%) | 498 (range: 208–1368) | ▲ Camp: Jordan, Kenya<br>▲ Village: Pakistan, Yemen<br>▲ District: Yemen |
| MAM (n=9)[10 19 20 25 28 34 36 38 44] (Z-score <−2 to <−3 SD) | ▲ IDPs: Afghanistan, Bosnia-Herzegovina, Sierra Leone, South Sudan, Sudan, Ukraine<br>▲ Refugees: Algeria, Jordan, Pakistan<br>▲ Not specified: East Timor, Yemen | ▲ Camp: Algeria, Ukraine<br>▲ Community: Bosnia-Herzegovina, Jordan, Pakistan, South Sudan, Sudan, Yemen | 4.0% (range: 0.27%–25%) | 563 (range: 111–250000) | ▲ Camp: Algeria<br>▲ Village: Pakistan<br>▲ District: Yemen<br>▲ Governorates: Jordan |

Continued

**Table 1** Continued

| Indicators | Target population: country | Setting | Estimates median (range) | Sample size median (range) | Scale |
|---|---|---|---|---|---|
| **SAM** (n=8)[18–20 23 28 32 38 44] (Z-score <–3 SD) | ▲ IDPs: South Sudan, Sudan<br>▲ Refugees: Jordan, Kenya, Pakistan<br>▲ Not specified: CAR, Yemen | ▲ Camp: Kenya<br>▲ Community: Jordan, Pakistan, South Sudan, Sudan, Yemen | 1.50% (range: 0.15%–5.3%) | 809 (range: 208–46 383) | ▲ Camp: Kenya<br>▲ Village: Pakistan, Yemen<br>▲ District: Yemen<br>▲ Governorates: Jordan |
| **Chronic malnutrition** | | | | | |
| **Stunting** (n=7)[10 18 20 23 25 28 38] | ▲ IDPs: Afghanistan, Sierra Leone, South Sudan<br>▲ Refugees: Algeria, Pakistan<br>▲ Not specified: East Timor, CAR, Yemen | ▲ Camp: Algeria<br>▲ Community: Pakistan, South Sudan, Yemen | 38.60% (range: 13.6%–53%) | 432 (range: 111–1055) | ▲ Camp: Algeria<br>▲ Village: Pakistan, Yemen<br>▲ District: Yemen |
| **Overweight** (n=1)[38] | ▲ Refugees: Pakistan | ▲ Community | 18.1% | 1055 | ▲ Village |
| **Anaemia** (n=1)[38] | ▲ Refugees: Pakistan | ▲ Community | 23.2% | 1055 | ▲ Village |
| **Other indicators** | | | | | |
| **Diarrhoea prevalence** (n=3)[10 24 28] | ▲ IDPs: Guatemala, Pakistan<br>▲ Not specified: Yemen | ▲ Camps, hospital and mobile clinics: Pakistan<br>▲ Community: Yemen | 30% (range: 28.9%–73.8%) | 303 (range: 42–563) | ▲ Healthcare facilities: Pakistan<br>▲ District: Yemen |
| **Mortality due to diarrhoea** (n=3)[10 34 38] | ▲ IDPs: Bosnia-Herzegovina, Somalia<br>▲ Refugees: Iraq, Nepal, Pakistan, Uganda, Zaire<br>▲ Residents: Eastern DRC | ▲ Camps: Southern Somalia<br>▲ Community: Bosnia-Herzegovina, Pakistan | 39% (range: 22.3%–87%) | 100 000 (range: 1055–3 000 000) | ▲ Village: Pakistan |

Note: GAM: Global Acute malnutrition; MAM: moderate acute malnutrition; SAM: severe acute malnutrition CAR, Central African Republic; DRC, Democratic Republic of the Congo; IDP, internally displaced person; IYCF, infant and young child feeding.

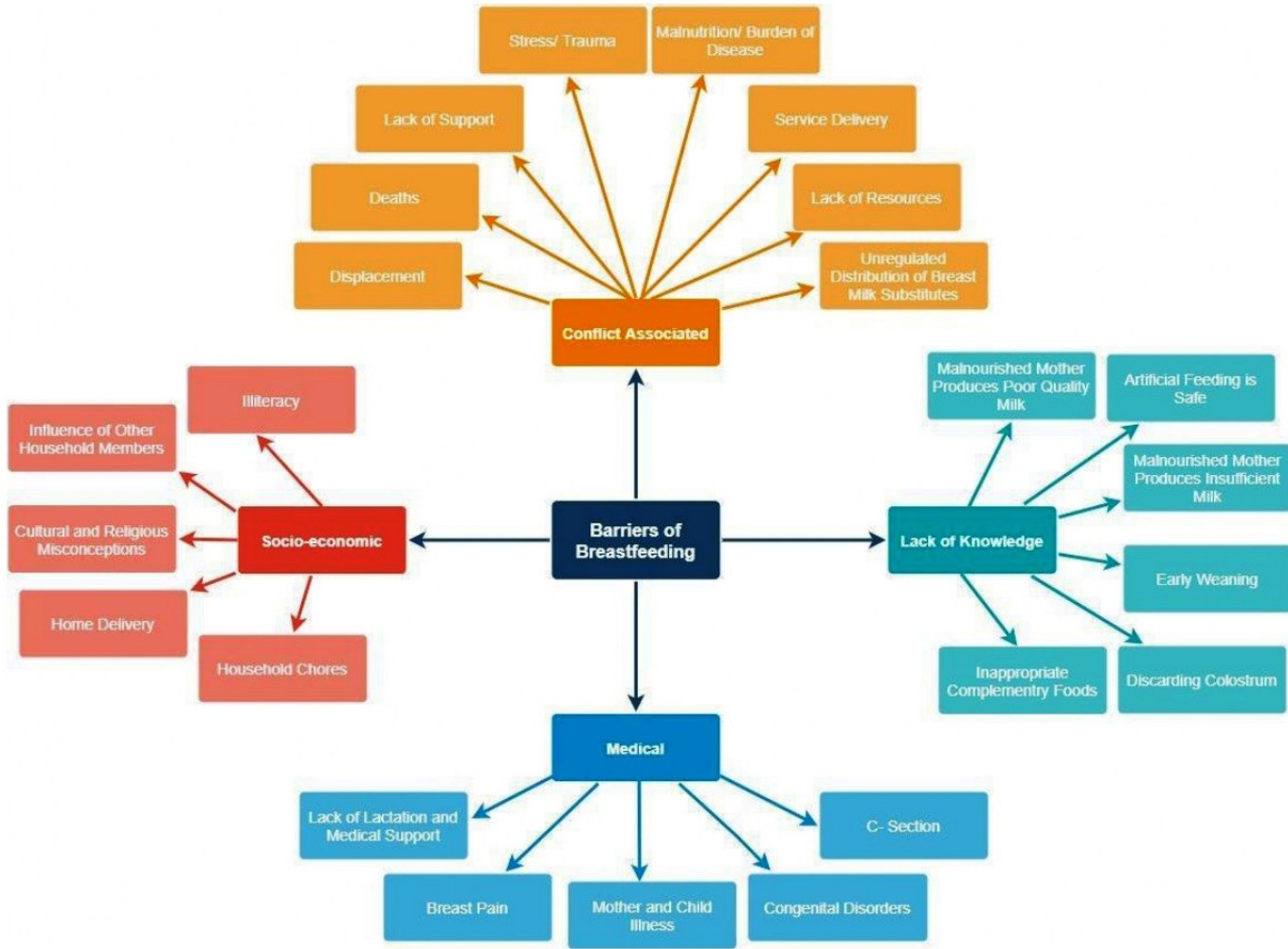

**Figure 3** Barriers to optimal breast feeding practices.

and wasting was low among children who were predominantly or exclusively breast fed (mean difference: 0.62, 95% CI 0.10 to 1.13 and 0.41, 95% CI −0.08 to 0.91, respectively).[25]

### Enablers/barriers
Eighteen studies reported on enablers/barriers to IYCF practices in conflict settings (figure 3).[13 14 19–22 25 27 29 33–37 40–43] Of these, 12 were reports[13 14 19–22 29 33 37 40 42 43] and 6 were cross sectional studies.[25 27 34–36 41] Seven studies were conducted in Europe,[27 33–37 43] five in Middle East,[13 14 19 21 22] three in Africa,[20 25 41] one in Asia,[29] one study was conducted in both Asia and Europe[40] and one failed to report.[41] Nine studies reported on refugees,[13 14 19 21 25 32 37 40 43] four on internally displaced persons (IDPs),[20 22 34 36] three on non-displaced residents[27 29 41] and two failed to report on the target population.[35 42] Nine studies were conducted in camps,[13 14 19 21 25 32 37 40 43] seven in community[20 22 27 29 34 36 41] and two did not report.[35 42]

### Conflict-related violence
Breast feeding declined significantly in regions with high levels of conflict-related violence relative to areas which were considered the 'safest areas' in Iraq.[29] A mother residing in dangerous areas of Iraq was 17.4% more likely to stop breast feeding compared with mothers living in the safe areas.[29] In Bosnia-Herzegovina, illness of mother/baby disrupted breast feeding and a few mothers made a personal decision not to breast feed.[35] Other reasons that negatively affected breast feeding practices in conflict affected areas were unavailability of trained healthcare professionals, and disruption in knowledge created by violent conflict.[19 29]

### Misconceptions
Ten studies reported the common misconception among refugees, media and humanitarian workers that mothers produce poor quality, insufficient milk due to malnourishment and stress induced as an impact of war.[19–22 29 33 34 37 40 42] These misconceptions led to mothers feeding undiluted animal milk to infants younger than 4 months.[20] In Bosnia-Herzegovina and Jordan, health professionals discouraged mothers from breast feeding due to significant maternal weight loss during conflict.[19 34] Breast feeding practices of mothers were affected by mental trauma from fighting and due to war-related casualties in male members of the family.[20 21 27 29 40]

In Bosnia-Herzegovina, infants were less likely to be breast fed for more than 4 months if they resided closer to the area of conflict and if the household was not receiving remittances from abroad.[27]

### Initiation of breast feeding

Studies conducted in Algeria, Jordan and Northern Uganda mentioned factors leading to delayed initiation of breast feeding by refugee mothers as home delivery, C-section, breast pain, discarding initial milk, illiteracy, lack of support from medical staff, previous experience, mother being ill and being the sole person responsible for decision of initiating breast feeding.[25 26 41] Mothers used pre-lacteal feeds before initiating breast feeding, as colostrum was believed to be of low nutritional value and was considered 'dirty' and harmful.[25 41] The report from Iraq and Greece mentioned that many mothers assumed that their milk was 'dirty' beacuse her previously born infant died while breast feeding, resulting in them refraining from breast feeding the next baby.[40]

### Introduction of complementary foods

Early introduction of complementary food and infant formula also led to suboptimal breast feeding practices.[13 21 33 35 42] They were introduced as early as 1–3 months because of the misconception that breastmilk was not sufficient for meeting the nutritional requirements of the infants.[21 35] Seven studies focusing mainly on refugees from conflict affected countries, explored reasons for mix feeding or fully artificial feeding infants.[13 19 21 33 35 40 42] Breast feeding and complementary feeding practices were influenced by religious and cultural determinants and frequent migrations.[29 33] Studies from Jordan and Bosnia-Herzegovina mentioned about the role of grandmothers in influencing breast feeding and complementary feeding practices.[19 35] Grandmothers often pressurised mothers to feed BMS (water and herbs) to infants and asked mothers to follow their traditional approach. Moreover, there was a cultural belief that formula milk is safer than breast milk and to introduce complementary feeding early.[19 40]

### Breast milk substitutes

Refugees in Lebanon and Greece considered BMS as necessary food for infants during emergencies and many mothers pursued mixed feeding because of the physician's prescription of infant formula.[13 33 42] The study on Syrian refugees in Lebanon stated that in children above 6 months, child refusal of food was due to inadequate knowledge regarding complementary feeding practices and the quantity of solid food that a young child should receive per day.[21] In Azraq refugee camp in Jordan, poor quality food and lack of iron-rich foods in the market was the reason for low consumption of iron-rich foods by the infants and young children.[26] In Bosnia-Herzegovina and Lebanon, mothers introduced tea, sugar, water, juice and infant formula in the feeding practice to spend less time breast feeding which negatively influenced the rate of EBF.[21 35] Two studies mentioned 'lack of milk'

as the reason for early weaning by mothers,[33 35] and many mothers wrongly thought that weaning cannot be reversed.[42] Another reason for suboptimal breast feeding practices, mentioned by four studies from Iraq, Greece, Lebanon and Jordan, was the heavy marketing of artificial feeding and big push from infant formula companies, which made mothers believe that it was better.[13 14 33 40]

### Programmes/interventions

Breast feeding in an emergency is known to be the safest way to protect infants and young children from an increased risk of infection and undernutrition.[14] To reduce the burden of disease and poor IYCF practices; several programmes were launched in conflict-affected countries by different organisations (WHO, international non-governmental organisations (NGOs) and local NGOs). The interventions were mostly based on the WHO guidelines to promote, protect and support appropriate IYCF practices.

We included 15 studies which reported on promotion of optimal IYCF practices in conflict settings (table 2).[10 13 14 18–24 32 35 40 43 44] Fourteen were reports[10 13 14 18–24 32 40 43 44] and one was a cross-sectional study.[35] These programmes were formulated based on literature review, document review and extensive discussions of stakeholders during conferences, through semistructured interviews (eg, open group discussions of mothers/caregivers), and knowledge, attitude and practices surveys. Six studies were conducted in Middle East,[13 14 19 21–23] four in Africa,[18 20 32 44] three in Europe,[10 35 43] one in Asia[24] and one study was conducted at more than one place, that is, Europe and Asia.[40] Eight of these focused on refugees,[10 13 14 19 21 32 40 43] three involved IDPs,[20 22 24] one reported on both hosts and IDPs[44] and three studies failed to report on it.[18 23 35] Six studies were conducted in refugee camps,[10 13 21 32 40 43] five in community,[18 20 22 23 44] three in both camps as well as in clinics[14 19 24] and one study failed to report on it.[35]

The specific interventions in these programmes included capacity building of healthcare staff, education and awareness activities for mothers, community mobilisation, provision of baby friendly spaces, lactation support services, complementary and safe artificial feeding support services, baby friendly hospital initiative (BFHI) and monitoring and control of BMS.[10 13 14 18–24 32 35 40 43 44] Five studies focused only on one intervention for promotion of optimal breast feeding practices,[22–24 35 43] while rest had a multipronged approach.[10 13 14 18–21 32 40 44] Most of the studies/programmes did not evaluate the programmes and its impact on IYCF practices or health and nutrition indicators. A follow-up survey after 3 months of intervention, assessed breast feeding practices among Syrian refugees in Jordan, which showed an increase in breast feeding knowledge from 49.5% in 2013 to 71% in community and 91.2% in health facility in 2014. However, no improvement in breast feeding practices was observed.[19] Similarly, in Greece, formula

**Table 2** Programmes/interventions to promote optimal IYCF practices

| Programme | Target population/countries | Setting | Health workforce involved | Programme/intervention details | Outcomes |
|---|---|---|---|---|---|
| Capacity building and programme-strengthening projects for health workers (n=7)[13 14 19 22 32 40 44] | ▲ IDPs: Sudan, Syria<br>▲ Refugees: Jordan, Kenya, Lebanon<br>▲ Not specified: Greece, Iraq | ▲ Camps: Jordan<br>▲ Fixed and mobile clinics: Jordan, Lebanon | ▲ Trained physician (doctors, nurses)<br>▲ Paramedic staff (midwives, TBAs)<br>▲ Community workers (facility and community based IYCF counsellors, local reproductive health worker) | ▲ Training sessions on IYCF<br>▲ Malnutrition screening and treatment<br>▲ Continuous follow-up and coordination | Coverage of training:<br>▲ In Lebanon and Syria: >190 doctors and health workers[27 29] |
| Education and awareness activities for mothers (n=7)[13 14 18 19 21 24 32] | ▲ IDPs: Pakistan<br>▲ Refugees: Jordan, Kenya, Lebanon<br>▲ Not specified: CAR | ▲ Camps: Jordan, Kenya, Lebanon, Pakistan<br>▲ Community: CAR, Lebanon<br>▲ Fixed and mobile clinics: Jordan, Pakistan<br>▲ Healthcare facility: Jordan Pakistan | ▲ Trained physician (lactation specialists)<br>▲ Community workers (nutrition officer, IYCF educator and a community mobiliser) | ▲ Counselling sessions on optimal IYCF-E practices<br>▲ Educational materials and counselling cards distributed<br>▲ Sensitisations on childcare practices and cooking demonstrations given to PLW<br>▲ Distribution of hygiene and baby kits (soaps, blankets, baby spoons and cups for children), and bangles for mothers | Coverage of IYCF counselling<br>▲ In Jordan: 30–40 mothers counselled/day[9]; 4690 PLWs and 919 mothers counselled in 10months[17]<br>▲ In Lebanon: 10000 mothers were counselled in 1 year[27]<br>▲ In CAR: 758/900 (84.2%) PLW participated[16] |

Continued

**Table 2** Continued

| Programme | Target population/ countries | Setting | Health workforce involved | Programme/intervention details | Outcomes |
|---|---|---|---|---|---|
| Community networking and mobilisation (n=8)[13 14 19 20 23 32 40 44] | ▲ IDPs: South Sudan, Sudan<br>▲ Refugees: Jordan, Kenya, Lebanon<br>▲ Not specified: Greece, Iraq, Yemen | ▲ Camps: Jordan, Kenya<br>▲ Community: Lebanon, South Sudan, Sudan, Yemen | ▲ Community workers (refugee mothers, community leaders, 'leader mothers' (mothers trained by promotors to teach neighbour women), and 'neighbour women' (chosen by community) as community mobilisers) | ▲ Training of community mobilisers by IYCF counsellors, educators, 'promotors', programme supervisors and IYCF coordinators on IYCF and the importance of exclusive breast feeding, nutrition<br>▲ Dissemination of messages among mothers through household visits, demonstrations and information sharing within care groups<br>▲ Screening and referring malnourished mothers | Coverage of health workers training:<br>▲ In South Sudan, 320 'leader mothers' trained IYCF[18]<br>Coverage of IYCF counselling<br>▲ In Jordan: 4977 PLWs and 31 485 caregivers in 10 months[17]<br>▲ In South Sudan: 320 neighbourhood groups, reaching 3832 women[18]<br>▲ In Yemen: 50%–60% increase in awareness of mothers/caregivers on nutritious food along with an increase in utilisation of local foods for preparing nutritious meals for infants[31]<br>Mass screening and SAM/MAM treatment<br>▲ In Yemen: 90% of children <2 years screened for SAM and MAM, 2563 children were treated for SAM, reduction in number of cases of SAM and MAM children; zero cases of SAM (MUAC<115 mm) in 13/68 model villages by the end of the project[31]<br>▲ In Sudan: 150 617 children screened for malnutrition[42]<br>▲ In Yemen: reduction in bottle feeding to almost zero[31] |

Continued

**Table 2** Continued

| Programme | Target population/ countries | Setting | Health workforce involved | Programme/intervention details | Outcomes |
|---|---|---|---|---|---|
| Mother baby friendly spaces and mother to mother support groups (n=10)[10 13 14 18–21 32 43 44] | ▲ IDPs: South Sudan, Sudan<br>▲ Refugees: Albania, Croatia, Jordan, Kenya, Lebanon<br>▲ Not specified: CAR | ▲ Camps: Albania, Kenya, Lebanon<br>▲ Caravans: Jordan<br>▲ Community: South Sudan, Sudan<br>▲ Containers: Croatia<br>▲ Primary healthcare centres: Lebanon | ▲ Trained Physician: paediatrician, lactation consultant<br>▲ Community workers: relief workers, psychosocial workers, paediatrician, lactation consultant, IYCF counsellors and community mobilisers | ▲ Construction of mother–baby friendly spaces, caravans and mother baby centre for counselling and 24 hours support<br>▲ Formation of mother to mother/caregiver support groups<br>▲ Distribution of high energy biscuit, a bottle of water, and shawls for privacy<br>▲ Distribution of food vouchers to mothers for nutritional security and psychological support<br>▲ Screening of children for malnutrition | Coverage of IYCF counselling in baby friendly spaces and mother-to-mother support groups:<br>▲ In Jordan: 15600 mothers in 18 months in Jordan[9]<br>▲ In Kenya: 581 facilitators trained in IYCF[36]<br>▲ In CAR: 199 mothers/ caregivers given psychosocial support[16]<br>Other outcomes:<br>▲ In Jordan: 120–150 mothers/ day visited baby friendly spaces in Jordan[9]<br>▲ In Jordan: 50 women attended support group gatherings[17]<br>▲ In Sudan: 14 272 infant/ mother pairs attended MtMSG[42]<br>▲ In Kenya: 713 MtMSG (2801 mothers/caregivers in MtMSG)[36] |
| Lactation support service (n=3)[10 13 19] | ▲ Refugees: Albania, Jordan, Lebanon, | ▲ Camps: Albania, Jordan<br>▲ Community: Lebanon | ▲ Trained physician (lactation specialists, obstetrician/ gynaecologist)<br>▲ Paramedic staff (midwives) | ▲ Assist mothers for re-lactation and in breast feeding difficulties (painful nursing, latching problems, low breast milk production and on correct positioning for feeding) | Coverage of counselling:<br>▲ In Lebanon: 3150 mothers in 6 months[27] |

Continued

**Table 2** Continued

| Programme | Target population/countries | Setting | Health workforce involved | Programme/intervention details | Outcomes |
|---|---|---|---|---|---|
| Baby friendly hospital initiative (n=2)[13 35] | ▲ Refugees: Lebanon<br>▲ Not specified: Bosnia-Herzegovina | ▲ Camps: Lebanon | ▲ Not specified | ▲ Labelling of maternity wards as 'baby-friendly' to support breast feeding<br>▲ Capacity building of health workforce<br>▲ Provision of tools and equipment to support breast feeding, reducing use of BMS | In Bosnia-Herzegovina, from 1997 to 1999:<br>▲ Predominant breast feeding increased from 64.3% to 77.3%<br>▲ Continued breast feeding at 2 years increased from 8.5% to 40.7% |
| Breast milk substitutes (n=5)[13 14 19 21 40] | ▲ Refugees: Jordan, Lebanon<br>▲ Not specified: Greece, Iraq | ▲ Camps: Greece, Iraq, Jordan and Lebanon | ▲ Trained physician (lactation specialists and obstetrician/gynaecologist)<br>▲ Paramedic staff (IYCF midwife) | ▲ Training of healthcare staff and mothers on artificial feeding<br>▲ Counselling of mothers on importance of breast feeding, appropriate use of infant formula and on adverse effects of artificial feeding on infant's health<br>▲ Monitor and control the distribution of infant formula<br>▲ Provision of BMS supplies and kits (cups and clean water) for safe preparation of infant formula | ▲ In Lebanon: 50 infants were assisted with artificial feeding support[27]<br>▲ In Jordan: seven mothers received artificial milk supplies[9] |

BMS, breast milk substitutes; CAR, Central African Republic; EBF, exclusive breast feeding; IDPs, internally displaced persons; IYCF, infant and young child feeding; IYCF-E, infant and young child feeding in emergencies; MAM, moderate acute malnutrition; MtMSG, mother to mother support groups; PLWs, pregnant and lactating women; SAM, severe acute malnutrition; TBAs, traditional birth attendants.

use in the refugee camps decreased from 60% to 0% within 6 months, through their IYCF programme in emergencies.[40]

## Implementation guidelines

We included 30 implementation guidelines by different organisations in our review, mentioning key policies and operational guidelines to be followed during conflicts to ensure optimal IYCF practices.[1 2 6 26 30 33 42 45–68] Eight guidelines were by international NGOs,[1 30 33 42 46 51 59 67] nine by United Nations agencies,[6 47 50 53 55–57 65 66] and three by academic organisations,[48 49 64] while ten guidelines were formulated by collaboration between developmental partners and international NGOs.[2 26 52 54 58 60–63] These implementation guidelines were formulated after conducting random cluster surveys (including quantitative and qualitative analysis), meetings to gathering empirical evidence, past emergency experiences and technical guidance from community, stakeholders, development and implementation partners (online supplementary appendix table 1).

We included implementation guidelines from 1981 through 2018. Only one guideline was from 1981 to

1990,[50] seven were from 1991 to 2000,[30 47 51 53 54 66 67] 12 were from 2001 to 2010[2 6 42 46 48 49 52 55 56 62 63 68] and 10 were from 2011 to 2018.[1 26 32 45 57–59 61 64 65] These guidelines focused on refugees and IDPs affected by conflict. We summarised these operational guidelines from different organisations, to be used during conflict settings, using the components from 'WHO's guiding principles for feeding infants and young children during emergencies'[6] (figure 4).

### Breast feeding

The guidelines on protecting, promoting and supporting breast feeding were mentioned by 28 publications.[1 2 6 26 30 33 42 45–55 57–67] They recommend to practice EBF till 6 months of age, followed by frequent breast feeding until 2 years, along with complementary feeding to protect babies from infection, especially in crisis situations. There should be formation of practical guidelines, capacity building of healthcare staff and continuous flow of funds for the sustainability of programmes.

General population (including mothers) should be educated on breast feeding, its benefits, colostrum use, feeding methods and consequences of artificial feeding

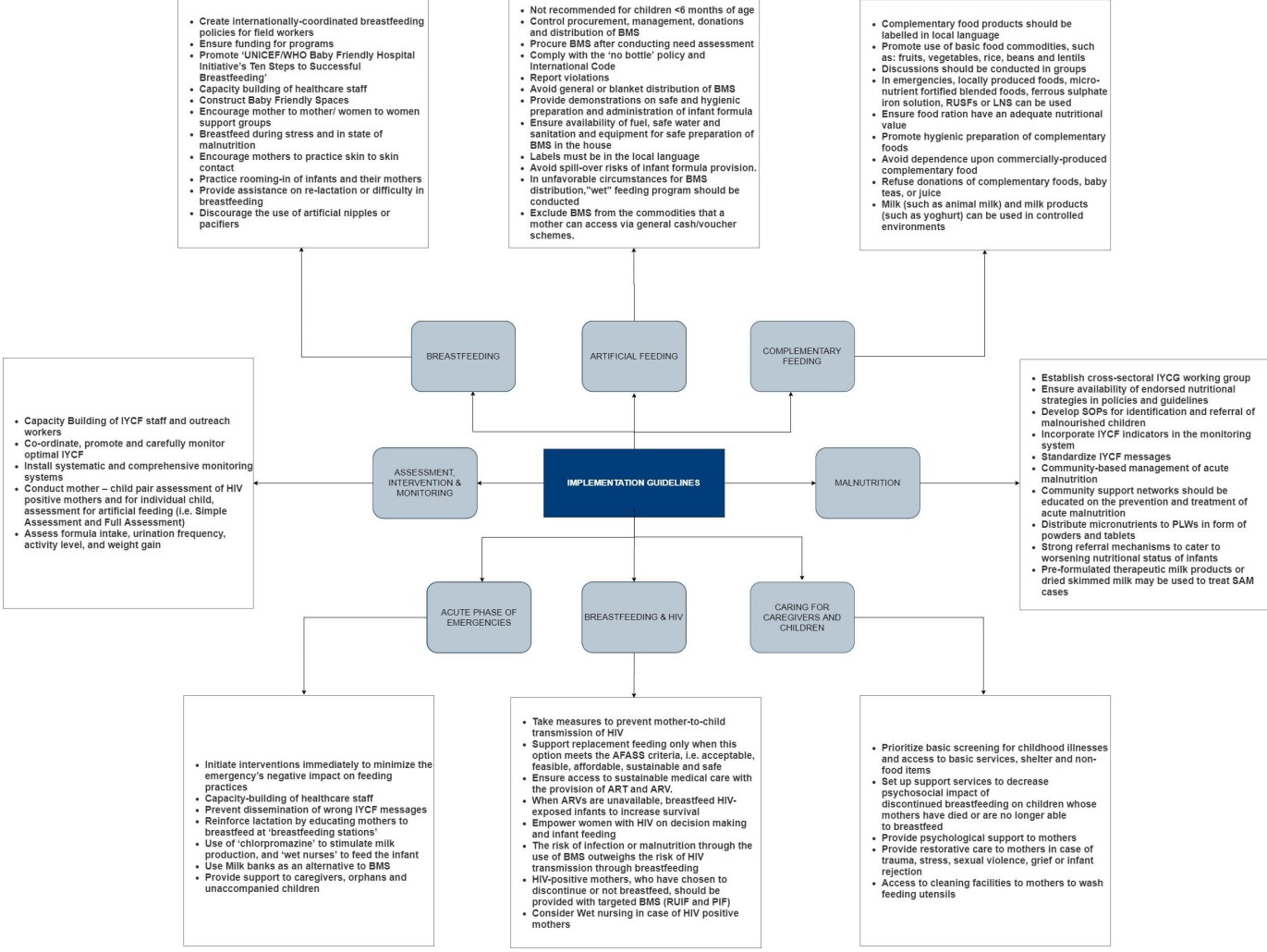

**Figure 4** Implementation guidelines.

based on 'UNICEF/WHO Baby Friendly Hospital's initiative's Ten Steps to breast feeding'. Moreover, mothers should be counselled not to stop breast feeding in emergency situations, sickness and when malnourished. They should be encouraged to practice skin-to-skin contact, initiate breast feeding within the first hour of birth, and use of pacifiers, artificial nipples and oestrogen containing contraceptive pills should be discouraged.

Guidelines also recommend formation of baby friendly spaces and mother to mother support groups to encourage breast feeding and privacy. There should be a system set in place for identification of newly arriving mothers and children, registration for rations and referral for immediate assistance. Mothers should be facilitated for re-lactation if separated from the baby or experiencing difficulties in breast feeding. When natural breast feeding is not possible, wet-nursing and the use of milk banks should be considered before providing infant formula or home-modified milk.

### Breast milk substitutes

Twenty-six guidelines reported on the use of BMS in emergency situations.[1 2 6 30 33 42 45–55 57–62 64 65 67] Donations, procurement, promotion (through advertisements and gifts) and distribution of BMS, bottles and teats should be strictly controlled and should comply with the International Code and World Health Assembly regulations, and all violations should be reported (see box 2). The distribution should be managed by a single designated agency and blanket distribution of BMS should be discouraged in emergency situations, especially where hygienic conditions cannot be ensured.

BMS are not recommended for children under 6 months of age and should only be distributed to infants who have no other viable breast milk options that is, orphans, severe maternal illness or malnutrition (based on established criteria—where distribution can be targeted, the supply chain is secure, and the conditions for safe preparation and use can be met), determined by a qualified health worker trained in IYCF. There should be training of workforce handling BMS and demonstrations should be given to mothers on proper and safe use of infant formula, followed by regular infant health and growth monitoring. Availability of fuel, safe water and equipment should be ensured before BMS distribution and mothers should be advised to use cups instead of bottles for feeding. Moreover, it should be ensured that distribution of BMS to the targeted infant continues for as long as the infant needs (at least 6 months).

Infant formula (generic, unbranded formula) should be distributed with labels in the local language to the infants in need. These labels should have clear instructions on its safe preparation, along with stating the importance of breast milk. Infant formula should not be excluded from the commodities that a mother can access via cash/voucher schemes, but it should have clear information on superiority of breast feeding. For infants under 6 months of age, the only suitable BMS is infant formula

---

### Box 2 Violation of IYCF-emergency guidelines in conflict settings (case studies)

**Case study 1: campaigning by Blédina company during Hezbollah–Israel conflict, 2006: Lebanon**

A marketing campaign by Bledina (production company of infant formula and complementary foods), after the war in 2006 in Beirut, offered mothers gift packages containing promotional leaflets. The attending paediatrician also gave mothers a card with the Blédina hotline. Moreover, free Blédina gift packages were also distributed at a health centre which had a high number of refugees, as a promotional campaign after the war, which was a violation of the IYCF-emergency guidelines.[12 46]

**Case study 2: civil unrest, 2002: East Timor**

Healthcare facilities in Dili distributed infant formula donated by a service organisation to assist infants who have been orphaned and mothers who are unable to breast feed, which was a violation of the guidelines because healthcare facilities should not distribute infant formula as it can be wrongly interpreted as them promoting the products, even if they are doing so to meet the needs of the targeted population.[46]

**Case study 3: Israel–Hezbollah conflict, 2006: Lebanon**

1. During the Lebanon conflict, an NGO violated the IYCF-emergency guidelines by distributing 1500 'baby kits', including infant formula and bottles, directly to displaced households, as well as to hospitals and local municipalities. Even after the conflict, the same NGO distributed 'village kits' containing infant formula (25 boxes containing 24 cans each) and baby food (80 units) along with other items to each village municipality.[12 46]
2. Another violation was the donation of infant formula to the healthcare facilities by an international NGO during the Lebanon conflict.[12]
3. Moreover, one local NGO distributed the infant formula among the displaced population, without clearly specifying on the tins that it should only be used on the advice of a health worker thus violating the labelling requirements, and promoting infant formula use.[12]
4. Another violation occurred when single tins/samples of infant formula were distributed to mothers by health workers, without the undertaking that the infant formula supplies would continue for as long as the concerned infant needs them.[12]

**Case study 4: Israel–Hezbollah conflict, 2006: Lebanon**

1. The formulas received by Lebanese government organisations and NGOs had labels written in English and/or Greek, instead of the local language, Arabic.[12 46]
2. One local NGO distributed infant formula, advertising and promoting artificial feeding, which was in violation of labelling requirements.[12 46]

IYCF, infant and young child feeding; NGO, non-government organisation.

---

and condensed milk should not be used for infant feeding and home-modified milk should be the last resort.

Relief workers should ensure that milk products are received and distributed in a dry form and dried milk products are distributed only when premixed with a milled staple food and should not be distributed as a single commodity. Moreover, it should be ensured that dried skimmed milk is not given to infants and for older children, it should be given after fortifying it with vitamin A. Relief and healthcare workers should counsel mothers to avoid baby juices and teas, and they should take appropriate measures to reduce spill-over by ensuring that feeding BMS to a minority of children does

not undermine breast feeding practices of the majority. In case of unfavourable circumstances for BMS distribution, an on-site supplementary 'wet' feeding programme should be conducted in closed spaces under supervision.

## Complementary feeding

Eighteen guidelines reported on complementary feeding in infants and young children 6–24 months of age.[1 6 26 42 45 48 49 51 54 55 57–59 61–65] For normal growth and development of infants and children (>6 months), easily digestible complementary foods should be started along with continuation of breast milk. Discussions should be conducted in groups, and mothers should be encouraged to increase the frequency and variety of complementary food with the growing age of child to meet their nutritional demands. Mothers should be encouraged to use locally produced, inexpensive complementary foods. However, in emergency situations, micronutrient fortified blended foods, ready-to-use supplementary foods, lipid based nutrient supplements or ferrous sulphate iron solution (iron drops) can be used depending on nutritional situation. Relief workers should ensure that complementary food products are labelled in local language with instructions on preparation and do not have the images of bottle feeding on them, and donations of complementary foods, baby teas or juices should be refused. For children over the age of 12 months, it is recommended that they should eat the same food as older children. If safe complementary foods are not available, mothers should be advised to continue breast feeding.

## Caring for caregivers and protecting children

Thirteen guidelines reported on caring for caregivers during emergencies[1 6 33 45 52 58 60–62 64–67] and nine reported on protecting children.[1 33 45 48 49 52 57–59 61 62 64 67] Guidelines recommend provision of psychological support and empowering mothers during crisis and advising them to continue breast feeding (via milk expression). It should be ensured that infants are screened for childhood illnesses and mothers of artificially fed infants have access to cleaning facilities for washing utensils for safe preparation of BMS as artificially fed infants are known to be at a greater risk of malnutrition, diarrhoea and chest infections. If an infant is ill with reduced appetite, mothers should continue breast feeding with smaller amounts and increased frequency. Relief staff should prioritise infants for re-lactation, re-establishment of EBF and BMS provision and associated support services.

## Malnutrition

Nine guidelines reported on management of malnutrition in infants and young children during emergencies.[1 6 52 54 55 58 60 65 67] Guidelines recommend continuous monitoring of nutritional status of mothers, infants and young children with the purpose of identifying, assessing, preventing and treating malnourished mothers and children. Malnutrition treatment and prevention programmes should incorporate and prioritise IYCF in

their agenda as the risk can be decreased with optimal IYCF practices. Education should be provided to community support networks on the prevention and treatment of acute malnutrition and cash/voucher programmes should be introduced. For prevention of malnutrition, micronutrients should be distributed to all pregnant and lactating women in form of powders and tablets. It is recommended to identify malnourished children through regular monitoring and they should be referred and admitted along with their mothers to a nutritional rehabilitation programme in case of severe malnourishment. In crisis situations, supplementary feeding should be the primary strategy for prevention and treatment of moderate acute malnutrition and pre-formulated therapeutic milk products or dried skimmed milk can be used to treat severe acute malnutrition.

## Acute phase of emergencies

Twelve guidelines reported on interventions to be undertaken during the acute phase of emergencies.[1 2 6 42 45 51 55 58 61 62 64 67] In case of an emergency, interventions should start immediately with focus on capacity building to improve IYCF practices, supporting caregivers and catering to nutritional needs of children in order to minimise the negative impact of emergency. In places with prior high infant formula use, appropriate interventions should be undertaken by relevant organisations to increase prevalence of appropriate IYCF practices and increase the rate of EBF. Measures should be taken to monitor BMS donations and distribution during the early phases of emergencies. Moreover, mothers should be educated and encouraged to breast feed every 2–3 hours at breast feeding stations scattered across the refugee sites. If mothers are experiencing difficulty in breast feeding, chlorpromazine can be used to stimulate milk production and wet-nursing and milk banks can also be used as an alternative to BMS.

## Assessment, intervention and monitoring

Fifteen guidelines mentioned assessment, intervention and monitoring during emergencies.[1 2 6 26 45 51 52 57–61 64 65 67] There should be a regular systematic monitoring to track BMS distribution and careful monitoring of optimal feeding and nutritional status of infants and young children. It is recommended to conduct weight monitoring, assess intake, urination frequency and activity level for those receiving BMS. Healthcare staff should use qualitative as well as quantitative methods to gather data regarding pre-crisis practices, demographics, morbidity, mortality, malnutrition and current IYCF practices.

## Breast feeding and HIV

Ten guidelines reported on breast feeding in HIV situations.[1 45 47 55–58 60 63 67] Guidelines recommend that appropriate measures should be undertaken to prevent mother-to-child transmission of HIV, and improve child survival from HIV. If the mother's HIV status is negative or unknown (or HIV testing is not available), she should

be advised to continue age appropriate breast feeding and replacement feeding should only be supported if is acceptable, feasible, affordable, sustainable and safe. For sustainable access to medical care, HIV positive mothers should be provided with antiretroviral treatment and even in case of unavailability, breast feeding should be continued.

If HIV-positive mothers choose not to breast feed the infant, appropriate BMS should be provided along with counselling on the risks of mixed feeding and artificial feeding. Wet nursing should also be considered in cases of HIV-positive mothers and wet nurse should be counselled on prevention of disease transmission. IYCF staff should make supportive arrangements for HIV-positive mothers to build confidence, reduce isolation, encourage age appropriate feeding and educate the family members to provide full support to the mothers and conduct regular follow-ups. Furthermore, activities related to prevention/elimination of mother-to-child transmission should be regularly done as part of nutritional interventions and the measures taken by IYCF staff should be sensitive.

## DISCUSSION

This review included a total of 56 primary studies, comprising 11 published articles and 45 reports from grey literature. The included studies covered IYCF practices, programmes and guidelines for the countries affected by armed conflict. The review shows that the coverage of IYCF practices are low in conflict settings; with only half of the children receiving early initiation of breast feeding and only a quarter being exclusively breast fed. The other IYCF indicators are also not encouraging, with high rates of bottle feeding.

The low IYCF indicators can be attributed to a multitude of factors in conflict settings which may also exist in non-conflict settings, though armed conflicts tend to aggravate and amplify these and there are also factors unique to conflict settings. Displacement, stress, maternal malnourishment, lack of awareness and unavailability of trained healthcare professionals, are all reasons contributing to poor IYCF. The death of the male members within the family also poses an additional barrier, as apart from increased maternal mental stress, it also leads to additional maternal responsibilities. This in turn may compromise the mothers' ability to provide attention to their children. The review highlights certain misconceptions in the community that contribute to the misguided belief that maternal malnourishment and stress lead to insufficient quantity and quality of breast milk and may stimulate the undesired use of BMS. Additionally, in some instances health workers due to their lack of awareness and knowledge, advocated for and prescribed BMS resulting in the unregulated marketing, provision and distribution of BMS.

The evidence from this review suggests that there is a need to enhance the capacity of health workers and to improve their communication with the community by using various channels including pictorials, videos or face to face meetings. Though various programmes have been initiated in conflict settings[69]; unfortunately, none of them have been formally evaluated to gauge the impact of these different approaches on IYCF indicators. As discussed in the review, the major evidence and recommendations derived from the experiences of the various implementation agencies working in such contexts rather than from scientific evaluation of programmes. These guidelines suggest that first and foremost, the importance of IYCF should be underscored and it should be a top priority for improving the health of children in conflict settings. There should be early dissemination of policies to all concerned agencies and healthcare workers and steps should be taken to register women and mothers in camps, as IDPs or as residents of conflict-inflicted areas. The educational approaches should be drafted with specific messages to alleviate the context-specific misconceptions within the community. There is evidence from various communications platforms which could be used to spread relevant messages including women support groups, involving prominent community members, designing pictorials, brochures or videos. The need for designated places like feeding tents was also emphasised as these could provide personal spaces for women to feed their children, seek support from peers and also use them as avenues for skin to skin care for preterm and low birth weight infants. The provision of clean hygienic utensils and safe drinking water should also be ensured for preparing complementary feeds. The guidelines also emphasise the need for strengthening BFHI in the functioning health facilities. Apart from access to required healthcare, mothers should also be provided with lactation and psychological support. The factors identified which negatively affect IYCF practices include, high turnover rate of health workers, lack of funds, poor multi-sectoral coordination, poor monitoring and evaluation system, more focus on malnutrition treatment than prevention, strengthened marketing efforts of BMS by industries, and poor capacity at community level.

The additional approaches identified included 'wet nurse' and 'milk banks' which could be sought after confirmation from specialists solely for mothers who were unable to breast feed. BMS should be treated as the last resort[3] and efforts should be put in place that allow for stringent regulatory checks on BMS, nipples and pacifiers.[3 70] All of the commodities in a conflict setting should flow through a common medium and designated agency/agencies, who should be responsible for controlling donations and distributing appropriately labelled infant formula, and any violation to these should be reported with timely action taken.

The major strength of our review is that it systematically looks at various areas of IYCF which includes current IYCF practices, specific barriers and strategies to improve breast feeding and complementary feeding using evidence from implementation guidelines. The limitations of our review include restricted access to studies conducted by various NGOs/agencies as most of them do not report and inclusion of articles was limited to English language only and

only three studies were included from conflict-affected countries in the Asian region. Some of the included studies missed on reporting important information like reporting on study context (eg, year and scale of conflict/surveys), outcomes, process indicators and programme impact. Hence, there is a critical need of further research on the process of implementation, effectiveness of IYCF interventions and cost effectiveness of these interventions in conflict settings.

To ensure effective scale-up of interventions for promotion of IYCF in conflict settings, the emphasis of the stakeholders should be on advocacy and implementation of evidence-based context specific interventions. A multi-sectoral approach together with stringent monitoring and evaluation mechanisms should be in place with capacity building and accountability.

**Correction notice** The license type of the paper has changed from CC BY-NC to CC BY.

**Contributors** ZB and JKD conceived the idea of the review. ZAP developed the search strategy. AR, ZAP and FAS conducted the search and data extraction with specific inputs from JKD. AR, ZAP and JKD developed the first draft of the paper. ZB and JKD reviewed and finalised the final manuscript.

**Funding** The support of the Family Larsson-Rosenquist Foundation is greatly acknowledged. The funders had no role in the findings and writing of the manuscript.

**Competing interests** None declared.

**Patient and public involvement** Patients and/or the public were not involved in the design, or conduct, or reporting, or dissemination plans of this research.

**Patient consent for publication** Not required.

**Ethics approval** We obtained ethical approval of this study from the Ethical Review Committee of Aga Khan University, Karachi and the National Bioethics Committee, Pakistan.

**Provenance and peer review** Not commissioned; externally peer reviewed.

**Data availability statement** Data are available upon reasonable request. The datasets used for analysis in this study are available from the corresponding author on reasonable request.

**ORCID iDs**
Zahra A Padhani http://orcid.org/0000-0003-4777-7565
Jai K Das http://orcid.org/0000-0002-2966-7162
Zulfiqar Bhutta http://orcid.org/0000-0003-0637-599X

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
