## [Reviewer comments · BMJ Open]

ARTICLE DETAILS

TITLE (PROVISIONAL)	A systematic review of Infant and Young Child Feeding Practices in Conflict Areas: What the evidence advocates
AUTHORS	Das, Jai K.; Rabbani, Amna; Padhani, Zahra Ali; A. Siddiqui, Faareha; Bhutta, Zulfiqar

VERSION 1 – REVIEW

REVIEWER	Dr. Arun Gupta Breastfeeding Promotion Network of India (BPNI), INDIA
REVIEW RETURNED	12-Feb-2020

GENERAL COMMENTS	The study has dealt with an important subject. However, the manuscript needs amendments. Please see following comments: 1. Title: According to the methodology (page 7 line 128), it is a systematic review. However, the title does not reflect it. 2. Abstract: Background - Page 3 line 26 refers the study as a comprehensive review while the methodology section on page 7 line 128 says it is a systematic review. Kindly explain it. Results - Page 3 line 45-48, the text appears to be recommendation rather than reflecting results coming out of analysis of study data. Kindly clarify. Conclusions - on Page 3, line 52, please clarify which IYCF guidelines authors are referring to. Limitations and systematic review registration number are missing from the abstract. Kindly include them. 3. On page 5, some of the numbered point are general statements. It will be useful if only specific information is provided here. 4. Introduction: ● A section on objectives of the study with an explicit statement of questions being addressed with reference to participants, interventions, comparisons, outcomes, and study design (PICOS) for the systematic review is missing in the introduction. ● On page 6 line 93, Please replace 'optimal' with 'universal' as scaling up of breastfeeding to a near universal level prevents 823 000 annual deaths in children younger than 5 years ● on page 6 line 96, the text mentions 'emergency' together with 'conflict'. Word emergency is not reflected in the title, conceptual framework and methodology. Moreover, emergency is a wider term which includes natural disasters, man-made emergencies and complex emergencies. Conflicts can be included in the man-made emergencies. See: ENN guidelines at
--

https://www.enonline.net/attachments/3127/Ops-G_English_04Mar2019_WEB.pdf There is a need to clarify it.

- Breastfeeding and IYCF have been used interchangeably in the manuscript. Moreover, the manuscript largely provides information about breastfeeding and it talks less about the complementary feeding. Same is also true for the figure 1 and 3 on page 34 and 36. Is it possible to include more information about complementary feeding?

5. Methodology:

- The study pertains to IYCF practices in the conflict areas. Please explain why malnutrition is included in the methodology.
- On page 8 line 156, please put IBFAN in brackets.
- Review of existing international guidelines on the subject as one of the domain for the study does not match with the objective of the review.

6. Results:

- On page 9 line 207-210, text pertaining to malnutrition may be deleted as it does not match with the study objectives.
- On page 9 line 202-207, please mention that the data only reflect the studies included in the manuscript. The data given here may not be necessarily same as reported in the national health surveys of the study countries.
- Data on 'burden of disease' given on page 10, line 221 - 244 may be deleted as it does not match with the study objectives.
- In the list of specific interventions on page 12 line 320, please indicate if any study also mentioned implementation of the International Code of Marketing of BMS.
- Implementation of the guidelines: Review of existing international guidelines on the subject could be a separate article and may be removed from the manuscript. Rather, implementation of international guidelines in the conflict situations could have been included.

7. Discussion:

- page 18 line 496 - while mentioning the number of studies, please include the break-up as 11 published articles in the journals and 45 grey literature.
- In the first para on page 18, please compare the IYCF practice data to the global/regional data to contextualise the situation of IYCF practices in the reported countries.
- In the first para on page 18, reference to malnutrition and burden of disease may be deleted as this information does not gel with the objectives of the study.
- Most of the reasons cited by the included studies for suboptimal breastfeeding/iycf practices are general reasons which are true in any non-conflict situation also. Please mention this fact in the discussion. Also, point out if some specific reasons like heightened stress due to conflict is the cause.
- Statement on page 18 line 511-12 contradicts with the text on line 506-07.
- Text on lines 515 - 546 provides general recommendations synthesised from available guidelines on the subject. This is more like a narrative review rather than a systematic review. Please revise the text.
- Text on line 545-546 mentions RUSF and LNS which is beyond the scope of the study, hence may be deleted.

	 • Recommendations on page 19 line 560 - 563 are too broad and do not reflect results of the study.
--	--

REVIEWER	Arin A. Balalian Columbia University Mailman School of Public Health
REVIEW RETURNED	02-Mar-2020

GENERAL COMMENTS	Thank you for the opportunity to review this paper. The infant and young child feeding is a major issue facing the countries after the conflict, and thus this paper provides a great review of previous experiences in addressing this issue. ***General comment: Although the authors have mentioned that the focus of the paper is on Infant and young child feeding in general after conflict situation, it looks like they have instead mainly focused on breastfeeding and acute malnutrition. These are important; nonetheless, the children in post-conflict settings also face chronic malnutrition (such as stunting and overweight) and difficulty in receiving complementary food after the exclusive breastfeeding period, which I feel was somewhat not given considerable weight by the authors. Abstract Background While it looks like the objective of the paper is an infant and young child feeding in general, the authors have put the focus on breastfeeding only. Methodology Could the authors mention the name of databases they searched in the abstract? Results Breastfeeding is crucial in preventing malnutrition, but so is complementary feeding after 6 months. The results presented in the abstract should also include other findings not related to breastfeeding. Could the authors also present the findings regarding the interventions, such as how many were based on a literature review? how many interventions were evaluated, etc.... I think these are important findings of this article that could be presented in the results in the expense of removing some parts that solely focused on breastfeeding. “It is imperative... “These recommendations would more be suitable for the conclusion section. I am also not sure how feasible it is to create one single lead entity to implement and monitor the distribution of BMS globally... This is a very strong recommendation and perhaps would be more suitable for a commentary. Introduction The introduction was mainly focusing on breastfeeding. would the authors also discuss the importance of complementary feeding? It is important to define the term “emergency situation” for the purpose of this paper. The situations could be different based on the length and severity of the conflict, and the target population could suffer from different forms of malnutrition. One of the reasons that children’s are malnourished in post-conflict zones are the destruction of infrastructures such as WASH systems, roads, food distribution chain impairment, disruption of food production, Could the authors elaborate more about that? Objective of the paper: While the authors have mentioned IYCF in general, through the main text, they mainly focused on acute malnutrition. Maybe the
---

	objective of the paper should be refined to be more focused on breastfeeding practices and guidelines in post-conflict settings? PICOS is not very clear: in terms of population and the outcome. Conceptual framework: The term armed conflict should be replaced by post-armed conflict to reflect the fact that the authors included the studies after a cease-fire. Methodology Eligibility Criteria: It is unclear why the time-span of five years was chosen, as many children in post-conflict settings suffer from prolonged effects of war. The disruption of infrastructure results in prolonged disruption of IYCF, which itself can result in chronic malnutrition. Did the authors mean Google scholar? How the first 10 pages were sorted? Box1: Search Strategy: Why the keywords such as cyclone, drought were included in the search terms if they were not the focus of this paper? Search terms also mainly focused on breastfeeding.... This is incompatible with the objective of the study, which is IYCF in general. Terms such as chronic malnutrition, food supplements minimum dietary diversity were not included. Results In Table 1, The authors have found studies exploring "Malnutrition," "Chronic Malnutrition," and "Other indicators." However, they have not discussed these findings in the text. Burden of disease: The discussion of crude mortality during the conflict is unclear to me, while the focus of the paper is on malnutrition after the conflict. Line 233: Did the authors mean under 5 mortality rate due to diarrhea in afghan refugees in Pakistan? Enablers/Barriers: I would suggest the authors to create themes for this part , as the presentation of the results looks out of order. For example, themes such as misconceptions about breastfeeding, Marketing could be created. Programs interventions: Could the authors comment on the sustainability of IYCF programs (If available), and mention in Table 2 or in text which one of the programs were evaluated for IYCF indicators? Discussion Line 498: "sub-optimal in conflict settings.." post conflict? Line 517-518: Unclear... Could the authors mention how the suggestions in lines[521-525] are supported by their findings? How feasible is the recommendation to create a single designated agency? Don't the authors think that these suggestions would be more suitable for a commentary? In line 553, the authors mentioned about poo quality of the studies. However, no formal quality assessment tools were discussed through the paper? Could the authors further elaborate? Using contractions such as "couldn't"and "don't" is appropriate in formal English. I would suggest the authors to language edit the entire manuscript for Table 2: There were no interventions for complementary feeding among the children beyond 6 months of age, which could be explained by choice of search string terms. Please refer to the previous comments.
--	--

	Figure 2: 123 studies were removed, but the reason was not mentioned. Page 39: The table is not named, I surmise this is the WHO guidelines table. I think this would be more appropriate as a supplementary table (if it is not)
--	--

REVIEWER	Peter Herbison University of Otago New Zealand
REVIEW RETURNED	01-Apr-2020

GENERAL COMMENTS	The numbers in this manuscript are mostly presented in a reasonable way. The median and range is a reasonable way to present the data. But I think the word "range" should be written inside all the parentheses with a range. I am confused as to some of the results. For instance the "Introduction of solid, semi-solid or soft food" was presented as a percentage and I cannot understand how this could happen (this is not the only example of this). More detail must be given so that the results are understandable. There are other minor issues. The authors use the word "infant". Does this always mean the same age group or should it be specified. There is mention that Rwandan refugees were in Nepal. This seems to be strange. Is it true? The English is not bad, but could do with a little improvement here and there. Throughout the results the authors claim that studies "failed to report". This could mean many things. Does it mean that the study said that it measured something and then did not report it. Or that the information was not given in a way that could be used. Or that the study did not even study that particular aspect. But the most common issue with this paper is the use of abbreviations. There is an extensive glossary but the treatment of abbreviations in the manuscript is not consistent. Sometimes they are given in full the first time they are used and sometimes not. In addition many of the abbreviations are only used once or twice so it would make the paper clearer to have these in full and not abbreviate them at all. This needs to be tidied up.
---

VERSION 1 – AUTHOR RESPONSE

Reviewer: 1

Reviewer Name: Dr. Arun Gupta

Institution and Country: Breastfeeding Promotion Network of India (BPNI), INDIA Please state any competing interests or state 'None declared': None Declared

Please leave your comments for the authors below

The study has dealt with an important subject. However, the manuscript needs amendments. Please see following comments:

1. Title:

According to the methodology (page 7 line 128), it is a systematic review. However, the title does not reflect it.

Author's answer: The term "systematic review" has been added.

2. Abstract:

Background - Page 3 line 26 refers the study as a comprehensive review while the methodology section on page 7 line 128 says it is a systematic review. Kindly explain it.

Author's answer: Corrected. The term comprehensive review has been changed to systematic review.

Results - Page 3 line 45-48, the text appears to be recommendation rather than reflecting results coming out of analysis of study data. Kindly clarify.

Author's answer: Thanks, we have now corrected the text to reflect our findings and recommendations derived therefrom

Conclusions - on Page 3, line 52, please clarify which IYCF guidelines authors are referring to. Limitations and systematic review registration number are missing from the abstract. Kindly include them.

Author's answer: The source of IYCF guidelines have been added. There is no systematic review registration number.

3. On page 5, some of the numbered point are general statements. It will be useful if only specific information is provided here.

Author's answer: Modified as suggested

4. Introduction:

- A section on objectives of the study with an explicit statement of questions being addressed with reference to participants, interventions, comparisons, outcomes, and study design (PICOS) for the systematic review is missing in the introduction.

Author's answer: The last sentence in the background section now has this objective specified.

- On page 6 line 93, Please replace 'optimal' with 'universal' as scaling up of breastfeeding to a near universal level prevents 823 000 annual deaths in children younger than 5 years

Author's answer: thanks for pointing this. We have now replaced this statement.

- on page 6 line 96, the text mentions 'emergency' together with 'conflict'. Word emergency is not reflected in the title, conceptual framework and methodology. Moreover, emergency is a wider term which includes natural disasters, man-made emergencies and complex emergencies. Conflicts can be included in the man-made emergencies. See: ENN guidelines at

<https://eur03.safelinks.protection.outlook.com/?url=https%3A%2F%2Fwww.ennonline.net%2Fattachments%2F3127%2FOps->

[G_English_04Mar2019_WEB.pdf&data=02%7C01%7Cjai.das%40aku.edu%7C510231fbb3d543d8822408d7f8df9107%7Ca5d4252a02f94e6096f09733baae4919%7C0%7C1%7C637251511415588748&sdata=M3kMnH5UYuZSVzFTK8BolkDxpCKD2NrThWP3ARwl4jA%3D&reserved=0](https://eur03.safelinks.protection.outlook.com/?url=https%3A%2F%2Fwww.ennonline.net%2Fattachments%2F3127%2FOps-G_English_04Mar2019_WEB.pdf&data=02%7C01%7Cjai.das%40aku.edu%7C510231fbb3d543d8822408d7f8df9107%7Ca5d4252a02f94e6096f09733baae4919%7C0%7C1%7C637251511415588748&sdata=M3kMnH5UYuZSVzFTK8BolkDxpCKD2NrThWP3ARwl4jA%3D&reserved=0) There is a need to clarify it.

Author's answer: Term "emergency" has been removed. "Armed conflict" has been maintained.

- Breastfeeding and IYCF have been used interchangeably in the manuscript. Moreover, the manuscript largely provides information about breastfeeding and it talks less about the complementary feeding. Same is also true for the figure 1 and 3 on page 34 and 36. Is it possible to include more information about complementary feeding?

Author's answer: thanks for highlighting this, we have now added more details on complementary feeding. Also added this to the figures.

5. Methodology:

- The study pertains to IYCF practices in the conflict areas. Please explain why malnutrition is included in the methodology.

Author's answer: We had initially reported this as it was in the included studies. But we agree with your point and have now removed these sentences to align the paper with the stated objectives of the study.

- On page 8 line 156, please put IBFAN in brackets.

Author's answer: abbreviation removed

- Review of existing international guidelines on the subject as one of the domain for the study does not match with the objective of the review.

Author's answer: This was included as our objective as written in the last line of the background section 'and guidelines to improve IYCF practices' and also in the methodology section as 'We conducted a systematic review for the available published and grey literature, assessing looking at four domains including: epidemiology (coverage of key IYCF and malnutrition indicators), enablers/barriers (for recommended IYCF practices), interventions/programs (effectiveness in improving IYCF practices) and implementation guidelines to improve IYCF practices in conflict settings.' This is also highlighted in the conceptual framework and hence is retained in the revised version.

6. Results:

- On page 9 line 207-210, text pertaining to malnutrition may be deleted as it does not match with the study objectives.

Author's answer: we have now deleted this as suggested.

- On page 9 line 202-207, please mention that the data only reflect the studies included in the manuscript. The data given here may not be necessarily same as reported in the national health surveys of the study countries.

Author's answer: thanks, we have now clarified this limitation

- Data on 'burden of disease' given on page 10, line 221 - 244 may be deleted as it does not match with the study objectives.

Author's answer: We agree with this critique and have now deleted this accordingly.

- In the list of specific interventions on page 12 line 320, please indicate if any study also mentioned implementation of the International Code of Marketing of BMS.

Author's answer: This has also been added to the text

- Implementation of the guidelines: Review of existing international guidelines on the subject could be a separate article and may be removed from the manuscript. Rather, implementation of international guidelines in the conflict situations could have been included.

Author's answer: Thanks for this suggestion and we have made it clear that the section on the programs and interventions that were implemented in such contexts.

7. Discussion:

- page 18 line 496 - while mentioning the number of studies, please include the break-up as 11 published articles in the journals and 45 grey literature.

Author's answer: Added

- In the first para on page 18, please compare the IYCF practice data to the global/regional data to contextualise the situation of IYCF practices in the reported countries.

Author's answer: We respectfully disagree with the suggestion as the data that we have is from smaller studies and may not capture the whole conflict settings accurately. Hence comparing that with global data may not be possible.

- In the first para on page 18, reference to malnutrition and burden of disease may be deleted as this information does not gel with the objectives of the study.

Author's answer: we have now deleted this.

- Most of the reasons cited by the included studies for suboptimal breastfeeding/iycf practices are general reasons which are true in any non-conflict situation also. Please mention this fact in the discussion. Also, point out if some specific reasons like heightened stress due to conflict is the cause.

Author's answer: Thanks. This is a valid observation and we have now added this.

- Statement on page 18 line 511-12 contradicts with the text on line 506-07.

Author's answer: Corrected

- Text on lines 515 - 546 provides general recommendations synthesised from available guidelines on the subject. This is more like a narrative review rather than a systematic review. Please revise the text.

Author's answer: thanks for raising this, we have now revised the language to reflect that these are specifically a synthesis of the evidence from the review.

- Text on line 545-546 mentions RUSF and LNS which is beyond the scope of the study, hence may be deleted.

Author's answer: Deleted and replaced with "supplementary foods"

- Recommendations on page 19 line 560 - 563 are too broad and do not reflect results of the study.

Author's answer: Thanks, we have corrected this.

Reviewer: 2

Reviewer Name: Arin A. Balalian

Institution and Country: Columbia University Mailman School of Public Health Please state any competing interests or state 'None declared': None Declared

Please leave your comments for the authors below Thank you for the opportunity to review this paper. The infant and young child feeding is a major issue facing the countries after the conflict, and thus this paper provides a great review of previous experiences in addressing this issue.

***General comment:

Although the authors have mentioned that the focus of the paper is on Infant and young child feeding in general after conflict situation, it looks like they have instead mainly focused on breastfeeding and acute malnutrition. These are important; nonetheless, the children in post-conflict settings also face chronic malnutrition (such as stunting and overweight) and difficulty in receiving complementary food after the exclusive breastfeeding period, which I feel was somewhat not given considerable weight by the authors.

Author's answer: Thanks for reviewing the paper and for your constructive feedback, we have now added more details on complementary feeding.

Abstract

Background

While it looks like the objective of the paper is an infant and young child feeding in general, the authors have put the focus on breastfeeding only.

Author's answer: We have now added more details on complementary feeding, in the introduction, results and discussion.

Methodology

Could the authors mention the name of databases they searched in the abstract?

Author's answer: we have now added these.

Results

Breastfeeding is crucial in preventing malnutrition, but so is complementary feeding after 6 months. The results presented in the abstract should also include other findings not related to breastfeeding. Could the authors also present the findings regarding the interventions, such as how many were based on a literature review? how many interventions were evaluated, etc.... I think these are important findings of this article that could be presented in the results in the expense of removing some parts that solely focused on breastfeeding.

Author's answer: Thanks for pointing this, we have now added data on complementary feeding. We also added numbers of the studies included to the four domains of the results.

"It is imperative... "These recommendations would more be suitable for the conclusion section. I am also not sure how feasible it is to create one single lead entity to implement and monitor the distribution of BMS globally... This is a very strong recommendation and perhaps would be more suitable for a commentary.

Author's answer: Shifted to conclusion and the 'single entity' recommendation has been removed from the abstract.

Introduction

The introduction was mainly focusing on breastfeeding. would the authors also discuss the importance of complementary feeding?

Author's answer: Thanks for highlighting this, data on complementary feeding has been added.

It is important to define the term "emergency situation" for the purpose of this paper. The situations could be different based on the length and severity of the conflict, and the target population could suffer from different forms of malnutrition. One of the reasons that children's are malnourished in post-conflict zones are the destruction of infrastructures such as WASH systems, roads, food distribution chain impairment, disruption of food production, Could the authors elaborate more about that?

Author's answer: 'Emergency' has been removed with a more specific word of 'armed conflict/conflict'. We believe that a broader discussion of determinants is beyond the scope of our current review and has been addressed in a separate recent publication in BMJ Global Health (Als et al, 2020).

Objective of the paper:

While the authors have mentioned IYCF in general, through the main text, they mainly focused on acute malnutrition. Maybe the objective of the paper should be refined to be more focused on breastfeeding practices and guidelines in post-conflict settings?

PICOS is not very clear: in terms of population and the outcome.

Author's answer: Thanks for pointing this and as also raised by other reviewer, we have added the objective in the last of the background section and added it as a PICO format. We have also removed malnutrition and added more on complementary feeding.

Conceptual framework:

The term armed conflict should be replaced by post-armed conflict to reflect the fact that the authors included the studies after a cease-fire.

Author's answer: Included studies comprised of both: i. during conflict and ii. Within five years of the conflict's cessation. This has been further clarified in the 'eligibility criteria'

Methodology

Eligibility Criteria:

It is unclear why the time-span of five years was chosen, as many children in post-conflict settings suffer from prolonged effects of war. The disruption of infrastructure results in prolonged disruption of IYCF, which itself can result in chronic malnutrition.

Author's answer: The reviewer is absolutely right in highlighting that conflict can have long-lasting effects but there could also be conflict which are not at scale or prolonged and does not involve the whole country or region, hence we kept this as five years to be consistent across the different nature, duration and scale of conflict.

Did the authors mean Google scholar? How the first 10 pages were sorted?

Author's answer: We searched for additional data by entering the 'TITLES' of all included studies on Google Scholar and reviewing the first 10 pages to include any relevant missing studies.

Box1: Search Strategy:

Why the keywords such as cyclone, drought were included in the search terms if they were not the focus of this paper?

Search terms also mainly focused on breastfeeding.... This is incompatible with the objective of the study, which is IYCF in general. Terms such as chronic malnutrition, food supplements minimum dietary diversity were not included.

Author's answer: we have terms for complementary feeding in the search strategy like infant feed, complementary feed, IYCF, Infant and Young Child Feeding. And we included terms like cyclone, drought in the search to capture studies in which draught and other humanitarian emergencies occurred in addition to conflict. We just did not want to miss out on studies and have excluded studies which were not in conflict or post conflict (5 years) settings as also mention that we also excluded studies conducted in humanitarian emergencies apart from armed conflict.

Results

In Table 1, The authors have found studies exploring "Malnutrition," "Chronic Malnutrition," and "Other indicators." However, they have not discussed these findings in the text.

Author's answer: We have now removed this and also the malnutrition section from the draft.

Burden of disease:

The discussion of crude mortality during the conflict is unclear to me, while the focus of the paper is on malnutrition after the conflict.

Author's answer: We have now removed this as also suggested by other reviewer.

Line 233: Did the authors mean under 5 mortality rate due to diarrhea in afghan refugees in Pakistan?

Author's answer: This section has been removed as suggested.

Enablers/Barriers:

I would suggest the authors to create themes for this part , as the presentation of the results looks out of order. For example, themes such as misconceptions about breastfeeding, Marketing could be created.

Author's answer: thank you for this great suggestion, we have now created these themes.

Programs interventions:

Could the authors comment on the sustainability of IYCF programs (If available), and mention in Table 2 or in text which one of the programs were evaluated for IYCF indicators?

Author's answer: there is no data on sustainability and the last column of the table 2 states all the indicators that were assessed.

Discussion

Line 498: "sub-optimal in conflict settings.." post conflict?

Author's answer: removed sub-optimal. Included studies comprised of both: i. during conflict and ii. Within five years of the conflict's cessation. This has been further clarified in the 'eligibility criteria'

Line 517-518: Unclear...

Could the authors mention how the suggestions in lines [521-525] are supported by their findings?

Author's answer: this has been clarified

How feasible is the recommendation to create a single designated agency? Don't the authors think that these suggestions would be more suitable for a commentary?

Author's answer: Thanks, we have corrected this.

In line 553, the authors mentioned about poo quality of the studies. However, no formal quality assessment tools were discussed through the paper? Could the authors further elaborate?

Using contractions such as "couldn't" and "don't" is appropriate in formal English.

Author's answer: thanks, we have corrected this

I would suggest the authors to language edit the entire manuscript for Table 2: There were no interventions for complementary feeding among the children beyond 6 months of age, which could be explained by choice of search string terms. Please refer to the previous comments. Author's answer: we have extensively edited the draft and additions have been made to complementary feeding in the text.

Figure 2: 123 studies were removed, but the reason was not mentioned.

Author's answer: we have added this information as well.

Page 39: The table is not named, I surmise this is the WHO guidelines table. I think this would be more appropriate as a supplementary table (if it is not)

Author's answer: okay

Reviewer: 3

Reviewer Name: Peter Herbison

Institution and Country: University of Otago

New Zealand

Please state any competing interests or state 'None declared': None declared

Please leave your comments for the authors below

The numbers in this manuscript are mostly presented in a reasonable way. The median and range is a reasonable way to present the data. But I think the word "range" should be written inside all the parentheses with a range.

Author's answer: Thanks, we have added the word 'range'

I am confused as to some of the results. For instance the "Introduction of solid, semi-solid or soft food" was presented as a percentage and I cannot understand how this could happen (this is not the only example of this). More detail must be given so that the results are understandable.

Author's answer: This is the proportion of children and now we added this to the draft.

“proportion of children with appropriate introduction of solid, semi-solid or soft foods was 71.1% (range: 40.7% to 98.6%), proportion of children with minimum dietary diversity was 60.3% (range: 9.2% to 79.4%)”

There are other minor issues. The authors use the word "infant". Does this always mean the same age group or should it be specified. There is mention that Rwandan refugees were in Nepal. This seems to be strange. Is it true?

Author's answer: the word 'infant' has been used consistently. And when it includes older than one year age, then the term 'infants and young children' has been used.

Checked and corrected to "Bhutanese"

The English is not bad, but could do with a little improvement here and there

Author's answer: The entire paper has been proof-read for grammatical errors and improvement of English.

Throughout the results the authors claim that studies "failed to report". This could mean many things. Does it mean that the study said that it measured something and then did not report it. Or that the information was not given in a way that could be used. Or that the study did not even study that particular aspect.

Author's answer: this is what we mean as we are not clear whether the study measured it or not.

But the most common issue with this paper is the use of abbreviations. There is an extensive glossary but the treatment of abbreviations in the manuscript is not consistent. Sometimes they are given in full the first time they are used and sometimes not. In addition many of the abbreviations are only used once or twice so it would make the paper clearer to have these in full and not abbreviate them at all. This needs to be tidied up.

Author's answer: Thanks, we have removed all the additional acronyms.

VERSION 2 – REVIEW

REVIEWER	Arin A. Balalian Columbia University
REVIEW RETURNED	03-Jul-2020

GENERAL COMMENTS	The author's have addressed all my comments. I have no further comments.
--

REVIEWER	Peter Herbison University of Otago New Zealand
REVIEW RETURNED	24-Jun-2020

GENERAL COMMENTS	I have no further comments on this paper.
---